# Effects of Fermenting the Plant Fraction of a Complete Feed on the Growth Performance, Nutrient Utilization, Antioxidant Functions, Meat Quality, and Intestinal Microbiota of Broilers

**DOI:** 10.3390/ani12202870

**Published:** 2022-10-21

**Authors:** Haoxuan Sun, Da Chen, Huiyi Cai, Wenhuan Chang, Zedong Wang, Guohua Liu, Xuejuan Deng, Zhimin Chen

**Affiliations:** 1Key Laboratory for Feed Biotechnology of the Ministry of Agriculture and Rural Affairs, Institute of Feed Research, Chinese Academy of Agriculture Sciences, Beijing 100081, China; 2National Engineering Research Center of Biological Feed Development, Beijing 100081, China

**Keywords:** Arbor Acres broiler, intestinal microbiota, nutrient utilization, fermented feed, production

## Abstract

**Simple Summary:**

Fermented feed is an effective way to replace antibiotics in poultry. The purpose of this study was to explore the effects of different levels of fermented feeds on growth performance, nutrient utilization, antioxidant function, meat quality, and intestinal microorganisms of broilers. The current research results showed that adding 10% fermented feed significantly improved the growth performance in 1–21 days, and adding 5% fermented feed significantly improved the growth performance in 1–42 days. Adding 15% fermented feed significantly improved the metabolic rate of the birds in 19–21 days and significantly increased the monounsaturated fatty acid concentration in the chickens. Adding fermented feed significantly reduced the cholesterol content in the chickens. In a word, adding 10% fermented feed significantly reduced the feed conversion ratio in 1–21 days and adding 5% fermented feed significantly improved the average daily gain and the average daily feed intake in 1–42 days. In addition, consuming fermented feed improved the meat quality of broilers.

**Abstract:**

We investigated the effects of fermenting the plant fraction of a solid complete feed (FPFF) on the growth performance, nutrient utilization, meat quality, antioxidant status, and intestinal microbiota of broiler chickens. The plant-based fraction of the complete feed was fermented using *Lactobacillus* and *Bacillus subtilis*. A total of 240, 1-day-old male Arbor Acres broilers were randomly allocated into four treatment groups, each comprised of six replicates. The groups were fed a corn–soybean meal-based diet (basic diet) or the same diet supplemented with 5%, 10%, or 15% FPFF for 6 weeks. As results, adding 10% fermented feed significantly improved the growth performance in 1–21 days, and adding 5% fermented feed significantly improved the growth performance in 1–42 days. Adding 15% fermented feed significantly improved the metabolic rate of the birds in 19–21 days and significantly increased the monounsaturated fatty acid concentration in the chickens. Adding fermented feed significantly reduced the cholesterol content in the chickens. In conclusion, adding 10% fermented feed significantly reduced the feed conversion ratio in 1–21 days and adding 5% fermented feed significantly improved the average daily gain and the average daily feed intake in 1–42 days. In addition, consuming fermented feed improved the meat quality of broilers.

## 1. Introduction

The use of fermented feed is increasing with the increasing challenges of maintaining the health of farmed animals and protecting the environment. Fermented feeds are a form of biological feed created by transforming the feed ingredients into microbial bacterial proteins, small bioactive peptides, amino acids, and active probiotics. Many studies have shown that feeding broilers fermented feed is advantageous, as it improves broiler production, nutrient utilization, immunity, and the intestinal microecological balance. The microorganisms that are commonly used to produce fermented feed include lactic acid bacteria, *bacilli*, *yeasts*, *molds* (*Aspergillus niger* and *Aspergillus oryzae*), and *Streptococcus faecalis*.

Lactic acid bacteria were the earliest microorganisms to be used and are the most widely used type. These bacteria grow under facultative aerobic or strict anaerobic conditions and ferment carbohydrates to produce lactic acid. The use of such bacteria to ferment feed produces a large amount of lactic acid, which improves the palatability of the feed, inhibits the growth of harmful bacteria in the intestines, and reduces the loss of dry matter (DM) and nutrients during feed storage [1,2,3]. *Bacilli* are aerobic bacteria that degrade proteins to form short peptides and polysaccharides to form low molecular-weight sugars, both of which are readily absorbable, and therefore improve the growth and feed utilization of animals. Many important biochemical reactions comprise the fermentation process, and these cannot be completed by a single microorganism. Instead, the joint action of two or more microorganisms is required. Therefore, mixed-strain fermentation has been developed [4], and numerous studies of mixed-strain fermentation have been performed and have generated encouraging results. However, because broiler chickens have relatively short intestines, the effective application of this approach to prepare broiler feed requires many studies. The studies conducted to date on the use of fermented feed for broiler chickens have largely used single-plant protein sources [5]. However, fermentation can cause the loss of nutrients, such as vitamins and amino acids, particularly synthetic amino acids [6]. As a result, fermenting the plant fraction alone would be preferable to fermenting the complete feed. We hypothesized that fermenting the plant fraction of a complete feed would promote the growth of broilers, and this study was designed to determine the growth performance and other related indices. In the present study, we generate a reference for the use of microbially fermented components in broiler feed. We used *Lactobacillus plantarum* and *Bacillus subtilis* to ferment the vegetable component of the complete broiler feed, combined the product with the remaining components of the complete feed in various proportions, and then fed these mixtures to broilers to study their effects on nutrient utilization, production, antioxidant function, and intestinal development.

## 2. Materials and Methods

### 2.1. Animal Welfare Statement

The authors confirm that the ethical policies of the journal, as described under the Author Guidelines, have been adhered to and an appropriate ethical review committee has approved this study. The authors confirm that they have adhered to European Union standards for the protection of animals used for scientific purposes and feed legislation.

This study was approved by the Animal Use Committee of the Feed Institute of the Chinese Academy of Agriculture Science and performed according to the guidelines for animal experiments of the National Institute of Animal Health (Statement no. AEC-CAAS-20181208).

### 2.2. Preparation of the Complete Feed Containing the Fermented Plant Fraction

The broilers were divided into four groups, in which the control group was fed the basic diet and no fermented feed was added. The other three groups were 5%, 10%, and 15% of the fermented feed added to the basic diet, respectively. The strains used in this experiment were *L. plantarum* (viable bacterial count > 1.0 × 10^9^ CFU/g) and *B. subtilis* (viable bacterial count > 1.0 × 10^9^ CFU/g), which were screened and preserved by the Key Laboratory for Feed Biotechnology of the Ministry of Agriculture and Rural Affairs. The corresponding bacterial liquid was simultaneously added to the plant fraction (corn, soybean meal, cottonseed meal, and rapeseed meal) in the broiler feed formula at the same time. The proportion of bacterial powder to the plant fraction was 0.1%. The plant fraction was packed in a breathing bag (PA/PE seven-layer co-extruded film, 50 cm × 50 cm, Chuangjia Packaging Material Co., Ltd. (Wenzhou, China)) and fermented for 4 days at 35 °C. The fermented plant fraction and premix were mixed according to the proportions shown in Table 1 to prepare the fermented feed. Distilled water was added to each treatment group to equalize the feed water content in each group. The fermented feeds were equal in energy and protein. Table 1 shows the ingredients and chemical composition of the basal diet (% DM basis).

### 2.3. Poultry Husbandry

The experiment was conducted at the Nankou Experimental Base of the Feed Research Institute of the Chinese Academy of Agricultural Science. Four diets were prepared, a basic diet and three experimental diets, which were created by supplementing the basic diet with 5%, 10%, or 15% of the microbially fermented feed. Table 1 shows the nutrient composition of the four feeds (% DM basis). Table 2 shows the nutrient composition of the plant fraction in broiler feed (% DM basis). All feed was fed to the broilers after granulation. A total of 240 1-day-old male healthy Arbor Acres broilers were obtained from the China Hebei Zhengda group and randomly allocated to each diet (four in total), with six replicates per diet and ten chickens per replicate.

### 2.4. Test Diet

The basic diet composition, nutritional content, the relevant nutritional tables of the feed for the broilers and the relevant nutritional tables of the plant fraction in broiler feed are shown in Table 1 and Table 2.

### 2.5. Experimental Design

The study was performed in two phases: the starter phase (days 1–21) and the finisher phase (days 22–42). A 16 h light/8 h dark regime was used throughout the experimental period. The birds were housed in floor pens (100 cm× 115 cm× 70 cm, straw litter) and had free access to feed and fresh water. The room temperature was maintained at 33–35 °C on days 0–3 and at 32–34 °C on days 4–7. We gradually lowered the temperature through the air conditioner. Temperature was set to 32 °C for the first week, then temperature was dropped by 2 °C each successive week until it reached 20 °C. The birds were vaccinated against Marek’s disease on day 1(Huadushihua Co., Ltd. (Beijing, China)), Newcastle disease, and infectious bronchitis on days 7 and 21, infectious bursitis on days 14 and 24, and Newcastle disease again on day 35. Hygiene management of the chicken house was performed routinely.

### 2.6. Growth Performance

The feed consumption and body weight of each replicate pen were recorded at 1, 21, and 42 days of age. Average daily gain (ADG), average daily feed intake (ADFI), and the feed conversion ratio (feed: body weight gain, FCR) were calculated based on these data.

### 2.7. Total Apparent Digestibility of the Feed Nutrients

After adding the feed materials and premix, TiO2 (0.4%) was added to the feed. Then the feed was mixed. Digestion and metabolism tests were carried out at 19–21 days of age and 40–42 days of age, and fresh fecal samples were collected regularly and repeatedly every morning. A 10% sulfuric acid solution was added to the fecal samples for nitrogen fixation, followed by drying in an oven at 65°C and rehydration for 24 h. The samples were pulverized through a 40-mesh sieve and stored at low temperatures. The titanium dioxide, crude protein (CP), and energy contents of the diets and feces were determined using a spectrophotometer, an automatic Kjeldahl nitrogen analyzer (KDY-9830, Tongrunyuan Co., Ltd. (Beijing, China)), and an oxygen bomb calorimeter (C2000, IKA Co., Ltd. (Guangzhou, China)), respectively.
The apparent metabolic rate for each feed nutrient (%) was calculated as = 100 – ((dietary content of marker/fecal content of marker) × (fecal content of nutrient/dietary content of nutrient)) × 100(1)
The apparent metabolic rate associated with each diet was calculated as = (fecal content of TiO_2_ − dietary content of TiO_2_)/fecal content of TiO_2_ × 100.(2)
The apparent utilization rate of total dietary energy was calculated as = (1 − (TiO_2_ content of the diet/fecal TiO_2_ content) × (fecal energy/dietary energy)) × 100.(3)
The apparent utilization rate of total dietary energy was calculated as = (1 − (dietary TiO_2_ content/fecal TiO_2_ content) × (fecal CP/dietary CP)) × 100.(4)
The apparent dietary metabolizable energy was calculated as = dietary energy − ((dietary TiO_2_ content × fecal energy)/fecal TiO_2_ content). The energy content was quantified in MJ/kg.(5)

### 2.8. Antioxidant Function

Before slaughtering, one broiler in each group was sampled from the thoracic artery, and the blood was collected at 5 mL/min at 3000 rpm for centrifugation 10 min, and the serum was collected. Serum total antioxidant capacity (T-AOC) and malonaldehyde (MDA) content of the chest muscle were determined as indices of antioxidant function, using the hydroxylamine method and thiobarbituric acid (TBA), respectively, according to the manufacturer’s instructions. (Nanjing Jiancheng Institute of Biological Engineering, Nanjing, China).

### 2.9. Cecal Microbial Composition

One broiler with average body weights, selected from each replicate pen, was sensitized by electronarcosis on day 42 of the experiment. The cecal chyme was collected in a frozen tube, labelled, snap frozen in liquid nitrogen, and quickly transferred to −80 °C, where it was stored until testing. The samples were thawed by transferring them from −80 °C to 4 °C. The total number of microorganisms, lactic acid bacteria, and *Escherichia coli* were determined as described by Sun [7].

### 2.10. Meat Quality

One randomly selected broiler from each replicate in each group was slaughtered, and a piece of breast muscle was used to measure meat quality.

Determination of the pH of the chest muscle was carried out 24th hours after the slaughter of the broilers. PH was measured by referring to the national standard “Determination of pH of Meat and Meat Products” protocol (GBT 9695.5-2008).

Cholesterol was determined in meat using the national standard “National Food Safety Standard-Determination of Cholesterol in Food” (GB5009.128-2016).

Fatty acids were determined by referring to the “Determination of Fatty Acids in Food Safety National Standard” (GB/T5009.1682016). An Agilent 6890 capillary gas chromatography column (Agilent Technologies Inc. (Wilmington, USA),30 m × 0.25 mm I.D, 0.20μm) was used at a detection temperature of 260 °C and a column temperature of 150 °C for 2 min, increased to 180 °C at 3 °C/min for 5 min, increased to 240 °C at 4 °C/min, and held for 18 min; the carrier gas was N_2_ (20 mL/min).

### 2.11. Statistical Analysis

The data were analyzed using the univariate method in the General Linear Model of SPSS 16.0 software (IBM Corp, Chicago, IL, USA), and the factors that reached the significance level of the F test (*p* < 0.05) were subjected to multiple comparisons using the LSD method.

## 3. Results

### 3.1. Growth Performance

No significant differences in the ADG or ADFI were observed between the groups during days 1–21 of the study. The 10% FPFF diet was associated with a lower feed conversion ratio (FCR) than the other diets during days 1–21 (*p* < 0.05). No significant differences in ADG, ADFI, or FCR were observed during days 22–42. The 5% FPFF diet was associated with higher ADG and ADFI compared with the other diets over 1–42 days of the trial (*p* < 0.05) (Table 3).

### 3.2. Total Apparent Digestibility of DM, CP, and GE

Judging from the nutrient utilization efficiency of the 19–21-day-old broilers, the total apparent digestibility of DM and GE associated with the 15% FPFF diet was higher than that associated with the basic diet or 5% FPFF diets (*p* < 0.05). The protein metabolic rates associated with the 10% FPFF and 15% FPFF diets tended to be higher than those associated with the basic diet (*p* < 0.1) (Table 4). However, no significant difference in nutrient utilization of the broilers was observed at the age of 40–42 days (Table 5).

### 3.3. Meat Quality

The level of supplementation of the diet with fermented feed significantly affected breast fat, cholesterol content, and pH. The cholesterol content of the breast muscle of the broilers that consumed FPFF was lower than that of birds that consumed the basic diet (*p* < 0.05). The pH of the 5% FPFF group was lower than that of the basic group (*p* < 0.05). Supplementing with the FPFF had no significant effect on the fatty acid content of the pectoral muscle, but it had a significant effect on fatty acid composition: consuming the 15% FPFF during the starter phase of the study reduced the saturated fatty acid concentration and increased the monounsaturated fatty acid concentration (*p* < 0.05) (Table 6 and Table 7).

### 3.4. Antioxidant Function

There was no significant difference in serum T-AOC content among treatments (*p* > 0.05). There was also no significant difference in serum MDA content among treatments (*p* > 0.05) (Table 8).

### 3.5. Intestinal Microbial Composition

The abundance of *E. coli* in the cecum of broilers fed the 15% FPFF tended to be lower than that of birds fed the basic diet, but the difference was not significant (Table 9).

## 4. Discussion

### 4.1. Growth Performance

The purpose of adding fermented feed components to a broiler diet is to improve production performance. Most studies have shown that the consumption of fermented feed has a positive effect on the growth performance of broilers. Current studies have focused on the effects of adding a single type of fermented plant material or fermenting the entire feed. Sugiharto showed that the effect of fermenting the plant component of the feed is superior to that of fermenting the entire feed. In the present experiment, the plant-based fraction of the complete feed was fermented and then added to the complete feed in set proportions [8]. As results, the 10% FPFF significantly reduced the FCR during the first 21 days of the study, but the ADG and ADFI of the broilers that consumed the 5% FPFF were significantly higher over the 42 days than those of birds feed the other diets. Thus, a small amount of fermented feed improved the growth performance of broilers.

Similar findings have been reported using diets that contained a proportion of fermented feed. For example, Chen showed that 10% fermented feed tends to increase the ADG and significantly reduce the FCR of broilers vs. 20% and 0% for fermented feed [9]. In addition, Cheng inoculated a sterile fermentation substrate with 4% *Bacillus licheniformis* to produce a fermentation product and added 2 g/kg to the diet of broilers, which significantly increased their ADG in the short term, and significantly reduced their FCR vs. the basic diet [3]. Furthermore, Shu used *B subtilis* and *Lactobacillus bulgaricus* to ferment feed to contain various proportions of fermented feed. The results showed that the fermented feed increased ADG and reduced feed consumption and FCR vs. the basic diet [10]. Ni used *Bacillus*, *yeast*, and lactic acid bacteria to ferment a mixture of cottonseed meal, sesame cake, and dried distillers grains with solubles for 96 h at 20 °C [11]. As a result, including 15% fermented feed increased ADG, and including 10% fermented feed reduced the feed cost most effectively. Zhang used *Geotrichum candidum*, *Candida tropicalis*, and *Brewer’s yeast* to ferment potato residue [12], which reduced the FCR of broilers and improved their utilization of feed protein. Adding 10% fermented feed to granulation significantly reduced the FCR during the starter term, compared with the basic diet and the diets containing other levels of replacements. This can be explained by the degradation of complex macromolecules (starch, cellulose, and protein) in the feed to form more easily digested and absorbed substances (monosaccharides, disaccharides, oligosaccharides, and amino acids), that can also be used by microorganisms. The ADFI associated with the 15% replacement level was significantly lower than that associated with the 5% replacement level, possibly because the microbes consumed a larger proportion of the feed nutrients, thereby reducing the amount supplied to the host for growth.

### 4.2. Total Apparent Digestibility of DM, CP, and GE

This study suggested that consumption of the fermented feed had a significant effect on the nutrient metabolism of broilers during the starter phase, and that the 15% supplementary level had the best effect, but including the fermented feed had no significant effects on nutrient metabolism during the finisher phase. In this study, fermented feed increased the nutrient utilization efficiency of 19–21-day-old broilers, but it did not improve the nutrient utilization efficiency of 40–42-day-old broilers. These results are consistent with previous studies. Diets fermented with *B**. subtilis* have been previously shown to reduce ammonia emissions by poultry by improving enzyme activity and nitrogen utilization [13]. In addition, *B**. subtilis* secretes proteases, amylase, and lipase. The secreted proteases degrade proteins to generate short peptides [14], thereby promoting the digestion and absorption of the feed by the broilers. Furthermore, *L plantarum* and *B subtilis* increase the content of short peptides (<600 Da) in the feed by nearly 62% [15], and the proportion of short peptides increases with the duration of fermentation [16]. The higher the digestive enzyme activity in the intestine, the higher the feed utilization rate and capacity for absorption [17]. Therefore, the reason why supplementing with fermented feed significantly affects nutrient metabolism in the starter term may be that the probiotics involved in fermentation promote digestion [18]. Indeed, the digestive system of the broilers was not fully developed during the starter phase, such that the endogenous levels of digestive enzymes would have been relatively low. What is more, in this study, 10% fermented feed only significantly increased the DM utilization efficiency of broilers, but had no significant effect on GE utilization efficiency and CP utilization efficiency. However, significant improvements were seen in DM and GE utilization efficiency for the 15% supplementary level fermented feed, but 15% fermented feed could not significantly improve FCR in broilers. It may be that 15% fermented feed promotes nutrient utilization, but microbial fermentation will consume nutrients, and too much addition will not lead to improvements in growth performance.

### 4.3. Meat Quality

The quality of the chicken meat consumed affects human health. In the present study, we showed that the fatty acids yielded by the feed were changed by fermentation. Stearic acid (C18:0), palmitic acid (C16:0), oleic acid (C18:1), and linoleic acid (C18:2) are the principal fatty acids in poultry meat, accounting for 85–95% of the total [19]. Unsaturated fatty acids are the principal mediators of the flavor of the muscle, and the effect of fermented feed on the fatty acid composition of broiler meat suggests that it has the potential to alter the flavor.

The effect of adding the fermented feed to the broiler diet on the fatty acid composition of the meat may be caused by metabolites produced during the fermentation process or yeast probiotics, but more detailed studies of the mechanisms are required. However, the effect on the taste of the meat may be mediated through the decomposition of the oxides generated by the double bonds of unsaturated fatty acids during the heating process to form carbonyl compounds, which have little aroma [20]. Wu showed that adding compound probiotic conditioners to a fattening pig diet increases the concentrations of monounsaturated and unsaturated fatty acids in the muscle, thereby improving pork quality [21]. In addition, Wang studied the effects of adding fermented components to the diet on the fat quality of growing and fattening pigs and found that fermented feed significantly increases the intramuscular fat content of pig longissimus dorsi and the oleic acid and total monounsaturated fatty acid concentrations. Reducing the linoleic acid and the total polyunsaturated fatty acid concentration of the muscle does not affect the concentrations of stearic acid or palmitic acid [22]. However, fermented feed significantly increases the concentrations of linoleic acid and polyunsaturated fatty acids in pig muscle, but does not affect the concentrations of stearic acid, palmitic acid, or oleic acid [23]. In this study, the cholesterol content in the breast muscle of the broilers fed the fermented feed was significantly lower than that of the basic diet group, and adding 15% fermented feed significantly reduced the concentration of saturated fatty acids and increased the concentration of monounsaturated fatty acids. This observation shows that adding fermented feed can improve the quality of chicken.

### 4.4. Antioxidant Function

The lipid content of broiler chicken meat is relatively high, and therefore lipid peroxidation is common. The serum concentrations of MDA and T-AOC reflect overall antioxidant status, and therefore health. In the present study, we showed that there is no significant difference in T-AOC and MDA, but adding 15% fermented feed to the broiler diet tended to reduce serum MDA levels and increase their T-AOC, suggesting an increase in overall antioxidant capacity. This may be because the presence of probiotics and the peptides produced by microbial degradation of proteins during fermentation improved the antioxidant properties of the feed. However, when *A**. niger* (ATCC 9142) was used to ferment grape pomace and the product was fed to broilers, the MDA concentration was unaffected [24]. Adding 5% or 10% fermented pomegranate residue to the feed of broilers affects their growth performance [25]. Niu added *Candida utilis* and *A**. niger* to ferment *Ginkgo biloba* leaves at ratios of 1.5, 2.5, 3.5, 4.5, and 5.5 g/kg to the broiler diet, and reported that all levels of supplementation significantly increased the T-AOC of the breast muscle and significantly reduced the MDA concentration of the leg muscle [26]. Another study showed that consuming fermented feed improved the antioxidant status of broilers [27], which is consistent with the results of this study.

### 4.5. Intestinal Microbial Composition

The normal function of the digestive system depends on the composition of the intestinal microbiota [28]. Previous studies have shown that adding fermented products containing probiotics to the diet affects the intestinal microbiota and immunity, and thereby improves the health and growth performance of poultry [29]. Hong showed that the consumption of fermented feed reduces the pH of the stomach and the number of coliforms in the gastrointestinal tract [30], and Sun showed that fermented feed limits the growth of pathogens such as *E. coli* [31]. Adding 10% or 15% fermented feed reduces the abundance of cecal *E**. coli* and *Salmonella* in broilers [32], and this was consistent with the results of the present study. However, Chen fed two-stage fermented feed generated using *B subtilis var. natto* N21 and S *cerevisiae* Y10 to broilers and reported no significant effects on the cecal abundance of lactic acid bacteria or coliform groups [33].

The improved intestinal microflora that is induced by feeding fermented feed is principally the result of a decrease in intestinal pH, an increase in the number of beneficial bacteria, and increases in the concentrations of lactic acid and acetic acid [5]. The fermented feed contains large quantities of lactic acid and other organic acids, which acidify the intestinal contents and stimulate the production of antibacterial bacteriocins [34]. In the present study, fewer bacteria were detected in broilers fed the 10% fermented feed than in the basic diet, but this difference was not significant because the number of live bacteria was reduced by granulation at a high temperature. Probiotics inhibit the growth of intestinal pathogens and the development of related diseases by producing antibacterial substances [35]. The fermentation process enriches the diet with short-chain fatty acids, vitamins, and enzymes, thereby improving the host intestinal environment and encouraging the growth of beneficial intestinal microbes [36]. However, the number of fermented products in the present study was insufficient to change the composition of the intestinal microbiota.

## 5. Conclusions

Taken together, adding 10% fermented feed significantly reduced FCR in 1–21 days, and adding 5% fermented feed significantly improved ADG and ADFI in 1–42 days. Adding 15% fermented feed improved the metabolic rate of 19–21-day-old broilers and significantly increased the monounsaturated fatty acid concentration in the chickens. Adding fermented feed significantly reduced the cholesterol content in the chickens. Thus, feeding fermented feed promoted the health of the broilers. This study provides a theoretical basis for applying plant fractions of fermented complete feed to broiler feed to enhance growth performance.

## Figures and Tables

**Table 1 animals-12-02870-t001:** Basic diet composition and nutritional content (air-dried basis) %.

Items	Starter Stage (1–21 Days)				Finisher Stage (22–42 Days)			
Corn	59.27	56.31	53.34	50.38	63.32	60.15	56.99	53.82
Soybean meal	29.01	27.56	26.11	24.66	22.22	21.11	20.00	18.89
Cottonseed meal	2.00	1.90	1.80	1.70	3.00	2.85	2.70	2.55
Rapeseed meal	2.00	1.90	1.80	1.70	3.00	2.85	2.70	2.55
Fermented feed	0	5.00	10.00	15.00	0	5.00	10.00	15.00
Vegetable oil	2.61	2.48	2.35	2.22	3.78	3.59	3.40	3.21
Salt	0.35	0.33	0.32	0.30	0.35	0.33	0.32	0.30
Dicalcium Phosphate	1.94	1.84	1.75	1.65	1.76	1.67	1.58	1.50
Stone powder	1.12	1.06	1.01	0.95	1.00	0.95	0.90	0.85
Lysine Hydrochloride	0.44	0.42	0.40	0.37	0.35	0.33	0.32	0.30
DL-methionine	0.23	0.22	0.21	0.20	0.20	0.19	0.18	0.17
L-threonine	0.13	0.12	0.12	0.11	0.16	0.15	0.14	0.14
L-tryptophan	-	-	-	-	0.01	0.01	0.01	0.01
Betaine	0.20	0.19	0.18	0.17	0.20	0.19	0.18	0.17
Choline Chloride	0.20	0.19	0.18	0.17	0.15	0.14	0.14	0.13
TiO_2_	0.04	0.04	0.04	0.04	0.04	0.04	0.04	0.04
Premix ^1^	0.13	0.12	0.12	0.11	0.13	0.12	0.12	0.11
Zeolite powder	0.37	0.35	0.33	0.31	0.37	0.35	0.33	0.31
Total	100	100	100	100	100	100	100	100
Nutritional level ^2^								
Metabolizable energy (Mcal/kg)	2972	3029	3048	3093	2910	2931	2953	2922
Crude protein (%)	19.96	20.16	20.35	20.55	18.56	18.68	18.79	18.91
Ether extract (%)	5.86	5.99	6.12	6.25	8.09	8.15	8.20	8.26
Crude fiber (%)	2.88	2.81	2.74	2.66	2.79	2.76	2.73	2.71
Crude ash (%)	4.76	4.85	4.92	5.00	4.09	4.13	4.16	4.20
Lysine (%)	1.250	1.263	1.287	1.325	1.050	1.057	1.070	1.090
Methionine (%)	0.500	0.505	0.515	0.530	0.450	0.453	0.459	0.467
Met + cystine (%)	0.921	0.930	0.948	0.976	0.844	0.849	0.860	0.876
Threonine (%)	0.760	0.768	0.783	0.806	0.720	0.725	0.734	0.747
Tryptophan (%)	0.200	0.202	0.206	0.212	0.180	0.181	0.183	0.187
Calcium (%)	1.000	1.010	1.030	1.060	0.900	0.906	0.917	0.934
Available phosphorus (%)	0.450	0.455	0.463	0.477	0.420	0.423	0.428	0.436

Note: ^1^ ① The premix provided the following per kg of diet for 1 to 21 days of age: VA, 8000 IU; VD_3_, 1000 IU; VE, 20 mg; VK_3_, 0.5 mg; VB_1_, 2.0 mg; VB_2_, 8 mg; VB_6_, 3.5 mg; VB_12_, 0.01 mg; Pantothenic acid, 10 mg; Niacin, 35 mg; Folic acid, 0.55 mg; Biotin, 0.18 mg; Choline chloride, 1300 mg; Cu, 8 mg; Fe, 80 mg; Zn, 80 mg; Mn, 80 mg; I, 0.7 mg; ② The premix provided the following per kg of diet for 22 to 42 days of age: VA, 6000 IU; VD_3_, 750 IU; VE, 10 mg; VK_3_, 0.5 mg; VB_1_, 2.0 mg; VB_2_, 5 mg; VB_6_, 3.0 mg; VB_12_, 0.01 mg; Pantothenic acid, 10 mg; Niacin, 30 mg; Folic acid, 0.55 mg; Biotin, 0.15 mg; Choline chloride, 1000 mg; Cu, 6.4 mg; Fe, 64 mg; Zn, 64 mg; Mn, 64 mg; I, 0.56 mg. ^2^ Crude protein, ether extract, crude fiber, crude ash, and metabolizable energy in nutrition levels are measured values. Other indexes in nutrition levels are calculated values.

**Table 2 animals-12-02870-t002:** Relevant nutritional tables of the plant fraction in broiler feed (air-dried basis) %.

	1 to 21 Days		22 to 42 Days	
Items	Feed	Fermented Feed	Feed	Fermented Feed
Crude Protein	21.63	25.87	20.27	22.84
Ether Extract	6.35	9.14	8.84	10.06
Crude Fiber	3.12	1.54	3.05	2.46
Crude Ash	5.16	6.83	4.47	5.28

**Table 3 animals-12-02870-t003:** The effects of adding fermented feed on the growth performance of broilers.

Items	0%	5%	10%	15%	SEM	*p*-Value
1–21 days						
ADG, g	41.08	41.07	42.37	40.45	1.97	0.202
ADFI, g	55.17	55.34	55.53	54.99	2.18	0.925
FCR	1.34 ^a^	1.35 ^a^	1.31 ^b^	1.36 ^a^	0.03	0.001
22–42 days						
ADG, g	84.82	88.27	82.47	83.12	12.59	0.700
ADFI, g	137.44	146.17	134.16	134.14	8.59	0.431
FCR	1.62	1.66	1.65	1.61	1.10	0.176
1–42 days						
ADG, g	62.95 ^b^	64.67 ^a^	62.42 ^b^	61.79 ^b^	4.37	0.026
ADFI, g	96.31 ^b^	100.76 ^a^	94.85 ^b^	94.57 ^b^	6.63	0.018
FCR	1.48	1.51	1.48	1.49	0.018	0.058

Note: ^a,b^ In the same row, values with different small letter superscripts indicate significant difference (*p* < 0.05); while those with the same letter indicate no significant difference (*p* > 0.05).The same as below. There are 6 replicate pens in the experiment, and one broiler is selected for each replicate pen. The same as below.

**Table 4 animals-12-02870-t004:** Effects of FPFF on tract apparent digestibility of DM, CP, and GE (19–21 days).

Items	0	5%	10%	15%	SEM	*p*-Value
DM%	68.89 ^b^	70.84 ^a,b^	71.72 ^a^	73.56 ^a^	1.95	0.010
CP%	61.49	61.91	65.51	67.87	3.08	0.093
GE%	72.13 ^b^	73.75 ^b^	74.47 ^b^	76.37 ^a^	1.85	0.022

^a,b^ In the same row, values with different small letter superscripts indicate significant difference (*p* < 0.05).

**Table 5 animals-12-02870-t005:** Effects of FPFF on tract apparent digestibility of DM, CP, and GE (40–42 days).

Items	0	5%	10%	15%	SEM	*p*-Value
DM%	69.53	70.09	69.18	69.20	3.18	0.719
CP%	60.60	61.93	50.03	63.17	5.06	0.729
GE%	72.33	73.05	72.56	72.30	3.22	0.780

**Table 6 animals-12-02870-t006:** Effects of adding fermented feed on the fatty acids, cholesterol, and pH of the breast muscle of broilers.

Items	0	5%	10%	15%	SEM	*p*-Value
Fatty acid	1.09	0.93	1.11	1.08	0.004	0.542
Cholesterol	0.45 ^a^	0.33 ^b^	0.37 ^b^	0.21 ^c^	0.080	0.007
pH	6.09 ^a^	5.78 ^b^	5.98 ^a,b^	5.98 ^a,b^	0.200	0.016

^a,b,c^ In the same row, values with different small letter superscripts indicate significant difference (*p* < 0.05).

**Table 7 animals-12-02870-t007:** The effect of adding fermented feed on the fatty acid composition of broiler breast muscle.

Items	0	5%	10%	15%	SEM	*p*-Value
C8:0	0.03	0.05	0.03	0.07	0.032	0.063
C10:0	0.11	0.07	0.04	0.19	0.115	0.207
C12:0	0.03	0.04	0.02	0.13	0.07	0.013
C14:0	0.55	0.56	0.51	0.35	0.124	0.001
C16:0	23.78	26.61	23.17	23.61	2.12	0.002
C18:0	10.75	12.68	11.64	13.13	1.807	0.226
C20:0	0.42	0.45	0.42	0.21	0.148	0.055
C22:0	0.31	0.3	0.23	0.15	0.1	0.012
C24:0	0.29	0.18	0.18	0.1	0.102	0.001
SFA	36.27 ^b^	40.92 ^a^	36.25 ^b^	37.94 ^b^	3.156	0.041
C16:1	1.8	2.28	1.79	1.61	0.459	0.026
C18:1	33.45	35.42	35	33.35	2.342	0.174
C20:1	0.41	0.45	0.5	0.51	0.08	0.021
C22:1	0.4	0.46	0.38	0.21	0.156	0.089
C24:1	0.35	0.24	0.22	0.15	0.112	0.001
MUFA	36.40 ^b^	38.85 ^a^	37.89 ^a^	35.83 ^b^	2.62	0.109
C18:2	13.76	12.42	14.44	13.86	1.3	0.06
C18:3	0.35	0.26	0.38	0.33	0.135	0.62
C20:2	1.31	1.04	1.27	1.32	0.305	0.054
C20:3	0.27	0.11	0.15	0.16	0.07	0.001
C20:4	8.58	4.96	7.09	6.93	2.58	0.005
C22:5	2.56	1.24	2.08	2.99	1.132	0.002
C22:6	0.4	0.15	0.35	0.53	0.254	0.009
PUFA	27.32 ^a^	20.22 ^b^	25.86 ^a,b^	26.24 ^a^	3.557	0.001

^a,b^ In the same row, values with different small letter superscripts indicate significant difference (*p* < 0.05); Saturated fatty acid (SFA) = C8:0 + C10:0 + C12:0 + C14:0 + C16:0 + C18:0 + C20:0 + C22:0 + C24:0; Monounsaturated fatty acid (MUFA) = C16:1 + C18:1 + C20:1 + C22:1 + C24:1; Polyunsaturated fatty acids (PUFA) = C18:2 + C18:3 + C20:2 + C20:3 + C20:4 + C22:5 + C22:6.

**Table 8 animals-12-02870-t008:** The effect of adding fermented feed on the antioxidant property indexes of broiler serum.

Items	0	5%	10%	15%	SEM	*p*-Value
T-AOC (mM)	0.67	0.64	0.61	0.80	0.052	0.273
MDA (nmol/mL)	4.00	4.06	4.51	2.89	0.162	0.453

**Table 9 animals-12-02870-t009:** The effect of fermented feed on the cecal microorganisms of broilers lg CFU/g.

Fermented Feed Ratio	Number of Bacteria	Number of Escherichia coli	Number of Lactic Acid Bacteria
0	11.17	7.65	7.17
5%	11.21	7.36	6.12
10%	11.14	7.28	6.92
15%	11.20	7.14	6.28
SEM	10.67	7.46	6.11
*p* value	0.232	0.235	0.102

## Data Availability

Data used in this study is available on request.

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
