# Peer review of "Effects of Fermenting the Plant Fraction of a Complete Feed on the Growth Performance, Nutrient Utilization, Antioxidant Functions, Meat Quality, and Intestinal Microbiota of Broilers"

_animals, 2022, doi:10.3390/ani12202870_

Round 1

Reviewer 1 Report

While the premise of this research is excellent and the topic of importance to the poultry industry, a final determination of the value of this manuscript cannot be made in its current form.

Major issues:

As you acknowledge in your manuscript (lines 260-263, and 277-281), the fermentation process can significantly alter the nutrient availability of the fermented feeds as components that would normally not be digestible by broilers are digested by the microbes and converted into digestible dietary components for broilers.  Therefore, the nutrient composition of the fermented plant fraction needs to be characterized, and then the nutrient composition of the individual diets needs to be calculated.  While table 1 presents the nutrient composition of the control diet the composition of the treatment diets is not presented and must be presented.

The abstract and conclusion sections need to be modified to reflect your actual statistically significant results.  Specifically line 27, growth performance was only improved by the 5% treatment, but not the other fermented treatments, and your statement implies they treatments containing fermented material improved growth performance.  The improvements in meat quality were also not consistent across the treatments and thus, the indication that quality was improved by utilizing fermented feed is misleading (line 28, line 370).  There were not significant differences in MDA and T-AOC presented in Table 8, so lines 28 -29 and 370 are not supported as written.  Finally, line 371 is not supported as the addition of the fermented feed component did not significantly alter measured cecal microorganism populations.

Lines 271-289, I think it might be important to add to the discussion that while significant improvements were seen in dry matter and gross energy digestibility values for the 15% fermented product dietary inclusion, that this did not correlate with an improvement in the FCR for this treatment in the starter phase.

Blood collection and serum processing details are not detailed in the materials and methods, despite presenting serum T-AOC data.

Minor issues:

For the Experimental design section line 123, please indicate if the experiment was conducted in cages, or floor pens and the size of the cages.  The fact that feces was collected for digestibility studies implies that the birds were raised in wire cages, but this needs to be defined.

Line 134, please change each replicate were recorded to each replicate pen were recorded.

Line 138, indicate specifically when the titanium was added to the experimental diets.

Line 154, please indicate if it was one selected broiler per replicate pen.

Line 189, please delete the phrase about capital letter significance as there are no capital letter superscripts used.

For all tables please include a statement in the footnotes that n = 6 replicate pens and then indicate the number of birds sampled.

Line 221 and 226-227, although as discussed above, the differences are not significant, you would also want to delete the phrase “entire study period” as the measurements were only made at the conclusion of the experiment and not throughout the experimental period.

Author Response

Response to Reviewer 1:

  Reviewer 1 made many helpful comments and suggestions, and we thank the reviewer for them. We have taken all these comments and suggestions into account as follows:

【Comment 1】As you acknowledge in your manuscript (lines 260-263, and 277-281), the fermentation process can significantly alter the nutrient availability of the fermented feeds as components that would normally not be digestible by broilers are digested by the microbes and converted into digestible dietary components for broilers.  Therefore, the nutrient composition of the fermented plant fraction needs to be characterized, and then the nutrient composition of the individual diets needs to be calculated.  While table 1 presents the nutrient composition of the control diet the composition of the treatment diets is not presented and must be presented.

My response: The authors have given the nutritional composition of the feed in the basic diet group and other treatment groups in table 2. The nutrient composition of the fermented plant fraction was characterized in table 3. Because the plant fraction of fermented feed partially replaces the plant fraction of the basic diet, the composition of other treatment diets is consistent with that of the basic diet.

【Comment 2】The abstract and conclusion sections need to be modified to reflect your actual statistically significant results.  Specifically line 27, growth performance was only improved by the 5% treatment, but not the other fermented treatments, and your statement implies they treatments containing fermented material improved growth performance.  The improvements in meat quality were also not consistent across the treatments and thus, the indication that quality was improved by utilizing fermented feed is misleading (line 28, line 370).  There were not significant differences in MDA and T-AOC presented in Table 8, so lines 28 -29 and 370 are not supported as written.  Finally, line 371 is not supported as the addition of the fermented feed component did not significantly alter measured cecal microorganism populations.

My response: In response to your suggestion, the author has revised the full text, please check it.

【Comment 3】Lines 271-289, I think it might be important to add to the discussion that while significant improvements were seen in dry matter and gross energy digestibility values for the 15% fermented product dietary inclusion, that this did not correlate with an improvement in the FCR for this treatment in the starter phase.

My response: The author has modified it, please check it in 4.2.

【Comment 4】Blood collection and serum processing details are not detailed in the materials and methods, despite presenting serum T-AOC data.

My response: It has been modified, please check it in 2.8.

【Comment 5】For the Experimental design section line 123, please indicate if the experiment was conducted in cages, or floor pens and the size of the cages.  The fact that feces was collected for digestibility studies implies that the birds were raised in wire cages, but this needs to be defined.

My response: The author has modified it , The birds were housed in floor pens (100 × 115 × 70 cm3) and had free access to feed and fresh water, please check it in 2.5.

【Comment 6】Line 134, please change each replicate were recorded to each replicate pen were recorded.

My response:  The author has modified it , please check it in 2.6.

【Comment 7】Line 138, indicate specifically when the titanium was added to the experimental diets.

My response: The author has modified it , please check it in 2.7.

【Comment 8】Line 154, please indicate if it was one selected broiler per replicate pen.

My response: The author has modified it,One broiler with average body weights selected from per replicate pen,please check it in 2.9.

【Comment 9】Line 189, please delete the phrase about capital letter significance as there are no capital letter superscripts used.

My response: The author has modified it , please check it in 3.1.

【Comment 10】For all tables please include a statement in the footnotes that n = 6 replicate pens and then indicate the number of birds sampled.

My response: The author has modified it , please check it in table 4, the same as below.

【Comment 11】Line 221 and 226-227, although as discussed above, the differences are not significant, you would also want to delete the phrase “entire study period” as the measurements were only made at the conclusion of the experiment and not throughout the experimental period.

My response: The author has modified it , please check it in 3.4.

Thank you for all your kind comments!

Reviewer 2 Report

Dear Authors, 

1- The abstract. Please present your significant and outstanding results not insignificant ones. Remove lines 29 and 30 and rewrite lines 26-30. Also, there is no conclusion for the abstract. 

2- Please be consistent. Use either basic diet or control. This may cause confusion for readers. EX: Line 113 vs 100. Check the whole manuscript. 

3- Why using 8h darkness throughout the experiment? 

4- How the environmental temperature reduced? In the text the temp dropped to 20 by day 42. It looks temp was high (week 4 and 5) before the sampling day. please clarify in the text.

5- TiO2 %. Is the enough to measure the parameters? Mostly studies consider much higher levels.

6- Regarding the meat quality data. The data is limited, can the authors consider to add more data related to oxidation such as drip loss or color?

7- Its unclear what timepoint PH measured? (post-mortem)

8- Table 8. How authors can explain the MDA data. It looks an error during measuring MDA, where it increased for 5-10% vs control and suddenly dropped for 15%? Lines 221-222 needs to be revised and provide a clear statement regarding MDA results. 

9- Line 371. The following statement is untrue based on the obtained results where there was no significant changes among the treatments. "and intestinal microbial composition of the broilers. Thus, the 15% fermented 371 feed promoted the health of the broilers"

Regards, 

Author Response

Response to Reviewer 2:

  Reviewer 2 made many helpful comments and suggestions, and we thank the reviewer for them. We have taken all these comments and suggestions into account as follows:

【Comment 1】1- The abstract. Please present your significant and outstanding results not insignificant ones. Remove lines 29 and 30 and rewrite lines 26-30. Also, there is no conclusion for the abstract.

My response: The author has modified it , please check it in Abstract.

【Comment 2】2- Please be consistent. Use either basic diet or control. This may cause confusion for readers. EX: Line 113 vs 100. Check the whole manuscript.

My response: The author has revised it, and there are 13 errors in the full text, all of which have been changed to basic diet.

【Comment 3】3- Why using 8h darkness throughout the experiment?

My response: The European Union has formulated animal welfare requirements in the animal breeding link, and the overall requirements for all animal protection are 98/58/EC "General requirements for Animal Protection". For broilers (2007/43/EC), the requirements stipulate that regular lighting should be provided, ensuring at least 6 hours of dark time every day, so the author chooses 8 hours of dark time.

【Comment 4】4- How the environmental temperature reduced? In the text the temp dropped to 20 by day 42. It looks temp was high (week 4 and 5) before the sampling day. please clarify in the text.

My response: The author has modified it. We gradually lowered the temperature through the air conditioner. Temperature was set to 32 ºC for the first week, then  temperature was dropped by 2 ºC each successive week until it reached 20 ºC. please check it in 2.5.

【Comment 5】5- TiO2 %. Is the enough to measure the parameters? Mostly studies consider much higher levels.

My response: TiO2 is enough to measure this parameters. According to the AAFCO(Official Publication. Association of American Feed Control Offcials) and some paper (Determination of titanium dioxide in poultry feed and surimi by spectrophotometer,《Feed industry》,Volume 29, Issue 2,2008) , the addition of TiO2 in feed is generally 0.2%-0.5%.

【Comment 6】6- Regarding the meat quality data. The data is limited, can the authors consider to add more data related to oxidation such as drip loss or color?

My response: Thank you for your suggestion. More indicators in this aspect will be measured in future studies.

【Comment 7】7- Its unclear what timepoint PH measured? (post-mortem)

My response: The author has modified it, the author measured pH at 24th hours after slaughtering, please check it in 2.10.

【Comment 8】8- Table 8. How authors can explain the MDA data. It looks an error during measuring MDA, where it increased for 5-10% vs control and suddenly dropped for 15%? Lines 221-222 needs to be revised and provide a clear statement regarding MDA results.

My response: The author confirms that the results of MDA data are correct. The author examined the original data, and the author measured several parallel samples. There are great differences in blood indexes within the group, which leads to no significant difference in the results of statistical analysis. L221 has been modified, please see it in 3.4.

【Comment 9】9- Line 371. The following statement is untrue based on the obtained results where there was no significant changes among the treatments. "and intestinal microbial composition of the broilers. Thus, the 15% fermented 371 feed promoted the health of the broilers"

My response: The author has revised it to focus on the significant parts, please check it in Conclusion.

Thank you for all your kind comments!

Round 2

Reviewer 1 Report

Thank you for revising the manuscript.

While the addition of Table 3 in the revised manuscript provides more information about the fermented material and its influence on diet composition, I still think if the AME and essential amino acid concentrations are known on the fermented feed ingredient that they should be used to provide the calculated nutrient levels for each diet and this information should then be presented in Table 1. Based on the information provided in Table 3, with the ether extract and crude protein levels increasing in the fermented feed it is very likely that the amino acid levels and metabolizable energy are no longer equal across the diets and this may account for the observed results.

Minor issues:

Line 32, The 10% inclusion did not improve growth.

Lines 128 and 143, Given that this was a floor pen experiment please indicate what type of litter was used, and further explain how the fecal samples were collected without litter contamination.

Line 141, Include TiO2 in Table 1 as it appears that it was a component of the diets throughout the experiment.

Lines 226-228, Delete the first portion of this sentence and just say “There was no significant differences in serum MDA content among treatments.”

Author Response

Response to Reviewer 1:

  Reviewer 1 made many helpful comments and suggestions, and we thank the reviewer for them. We have taken all these comments and suggestions into account as follows:

【Comment 1】While the addition of Table 3 in the revised manuscript provides more information about the fermented material and its influence on diet composition, I still think if the AME and essential amino acid concentrations are known on the fermented feed ingredient that they should be used to provide the calculated nutrient levels for each diet and this information should then be presented in Table 1. Based on the information provided in Table 3, with the ether extract and crude protein levels increasing in the fermented feed it is very likely that the amino acid levels and metabolizable energy are no longer equal across the diets and this may account for the observed results.

My response: The nutrition levels of each treatment group have been listed. Please see table 1 for details. As can be seen from the table 1, the energy and protein of each treatment group are roughly the same.

【Comment 2】Line 32, The 10% inclusion did not improve growth.

My response: I have revised it carefully. Please see it in Simple Summary, Abstract, and Conclusion.

【Comment 3】Lines 128 and 143, Given that this was a floor pen experiment please indicate what type of litter was used, and further explain how the fecal samples were collected without litter contamination.

My response: The  type of litter which was used in the experiment was straw litter. The straw litter was soft, dry, no mildew and no pollution. During the 19-21 and 40-42 days of the experiment, the author collected a small amount of feces. During the collection, the author take the feces from above. The author examined the fecal samples carefully and blew off feathers, litter scum and other sundries with a hair dryer. I have revised it, Please see it in 2.5.

【Comment 4】Line 141, Include TiO2 in Table 1 as it appears that it was a component of the diets throughout the experiment.

My response: I have revised it carefully. Please see it in table 1.

【Comment 5】Lines 226-228, Delete the first portion of this sentence and just say “There was no significant differences in serum MDA content among treatments.”

My response: I have revised it carefully. Please see it in 3.4.

Thank you for all your kind comments!

Reviewer 2 Report

Dear Authors, 

Thank you for the revised version. I have no more comments. 

Author Response

Thank you for your kind comment.Best wish to you!

Round 3

Reviewer 1 Report

Thank you for revising the manuscript.

Line 21 and lines 34-35, the text still indicates that growth performance was improved from day 1 to day 21, but Table 3 and lines 191-192 indicate there was no differences in average daily gain among the treatments from day 1-21. This disagreement in the text needs to be resolved.  I think you mean to indicate in lines 21 and lines 34-35 that feed conversion was improved by the 10% level in the 1-21 day period.

The updated table 1 was critical to provide, as it now allows the reader to see the differences in essential amino acid levels between the diets.

Author Response

Reviewer 1 made many helpful comments and suggestions, and we thank the reviewer for them. We have taken all these comments and suggestions into account as follows:

【Comment 1】Line 21 and lines 34-35, the text still indicates that growth performance was improved from day 1 to day 21, but Table 3 and lines 191-192 indicate there was no differences in average daily gain among the treatments from day 1-21. This disagreement in the text needs to be resolved.  I think you mean to indicate in lines 21 and lines 34-35 that feed conversion was improved by the 10% level in the 1-21 day period.

My response: I have modified it. Please see it for details in Simple Summary, Abstract and Conclusion.

【Comment 2】The updated table 1 was critical to provide, as it now allows the reader to see the differences in essential amino acid levels between the diets.

My response: Thank you for your compliment and wish you all the best.

Thank you for all your kind comments!